# Maintaining Normothermia in Preterm Babies during Stabilisation with an Intact Umbilical Cord

**DOI:** 10.3390/children9010075

**Published:** 2022-01-05

**Authors:** Alexander James Cleator, Emma Coombe, Vasiliki Alexopoulou, Laura Levingston, Kathryn Evans, Jonathan Christopher Hurst, Charles William Yoxall

**Affiliations:** Liverpool Women’s Hospital, Liverpool L8 7SS, UK; acleator@doctors.org.uk (A.J.C.); Emma.coombe2@lwh.nhs.uk (E.C.); v.alexopoulou@nhs.net (V.A.); laura.levingston@lwh.nhs.uk (L.L.); Kathryn.evans@lwh.nhs.uk (K.E.); Jonathan.hurst@lwh.nhs.uk (J.C.H.)

**Keywords:** hypothermia, umbilical cord, preterm, resuscitation

## Abstract

Background: We had experienced an increase in admission hypothermia rates during implementation of deferred cord clamping (DCC) in our unit. Our objective was to reduce the number of babies with a gestation below 32 weeks who are hypothermic on admission, whilst practising DCC and providing delivery room cuddles (DRC). Method: A 12 month quality improvement project set, in a large Neonatal Intensive Care Unit, from January 2020 to December 2020. Monthly rates of admission hypothermia (<36.5 °C) for all eligible babies, were tracked prospectively. Each hypothermic baby was reviewed as part of a series of Plan, Do, Study Act (PDSA) cycles, to understand potential reasons and to develop solutions. Implementation of these solutions included the dissemination of the learning through a variety of methods. The main outcome measure was the proportion of babies who were hypothermic (<36.5 °C) on admission compared to the previous 12 months. Results: 130 babies with a gestation below 32 weeks were admitted during the study period. 90 babies (69.2%) had DCC and 79 babies (60%) received DRC. Compared to the preceding 12 months, the rate of hypothermia decreased from 25/109 (22.3%) to 13/130 (10%) (*p* = 0.017). Only 1 baby (0.8%) was admitted with a temperature below 36 °C and 12 babies (9.2%) were admitted with a temperature between 36 °C and 36.4 °C. Continued monitoring during the 3 months after the end of the project showed that the improvements were sustained with 0 cases of hypothermia in 33 consecutive admissions. Conclusions: It is possible to achieve low rates of admission hypothermia in preterm babies whilst providing DCC and DRC. Using a quality improvement approach with PDSA cycles is an effective method of changing clinical practice to improve outcomes.

## 1. Introduction

Preterm babies are particularly vulnerable to hypothermia after birth. Hypothermia is an independent risk factor for death [1,2] and a risk factor for the development of respiratory distress syndrome, intraventricular haemorrhage, late onset sepsis and severe neurodevelopmental impairment [3,4].

The benefits of Deferred Cord Clamping (DCC) at preterm birth are well documented, including a reduction in hospital mortality [5].

Across the UK, in 2018 only 5.2% of babies with a gestation below 32 weeks had a documented period of DCC lasting for 60 s or more (Data from Badger System, Clevermed, UK. Personal communication, CEO, Clevermed, Edinburgh, UK). Our unit has been an early adopter of DCC. We recruited babies into studies to evaluate the practicability of using the Lifestart trolley to facilitate DCC in 2012 and 2013 [6] and enrolled babies into a randomised controlled trial of DCC from 2013 to 2015 [7]. There was some practise of DCC subsequently, but following the publication of the trial evidence showing clear benefit, we undertook a quality improvement project and increased the rate of DCC for at least 120 s to 92% of eligible babies in 2018 and 2019 [8]. During that period of implementation, we noted an increase in the rate of admission hypothermia (<36.5 °C) to 20% from a previous stable baseline rate of 12% (Figure 1).

In addition to promoting DCC, we have also recently started providing delivery room cuddles (DRC) following stabilisation of the baby. Once the baby is stable, a brief period of physical contact with the parents takes place before transfer to the neonatal unit. This can be achieved whilst the baby is receiving respiratory support. Two recent randomised controlled trials have shown reductions in both post-partum depression and mother-infant bonding problems following DRC [9,10], although one trial [10] found lower temperatures one hour after birth in their DRC group.

The UK National Neonatal Audit Project (NNAP) standard relating to admission temperature for preterm babies states that 90% of babies born <32 weeks should have a temperature between 36.5 °C and 37.5 °C when first admitted to the Neonatal Intensive care Unit (NICU), i.e., requires avoidance of hyperthermia as well as hypothermia.

The aim of this project was to reduce the number of preterm babies who are hypothermic (temperature < 36.5 °C) on admission to our neonatal unit, whilst promoting and practising DCC and DRC. In view of the NNAP standard, rates of admission hyperthermia were also measured.

## 2. Materials and Methods

We aimed to provide stabilisation at birth with at least 2 min of DCC in all babies born before 32 weeks gestation. The period of 2 min was the same as that used in the Cord Pilot trial (7). DCC is provided in our unit using the LifeStart trolley (Inspirations Healthcare, Crawley, UK) in the delivery room. This is a resuscitation platform that facilitates stabilisation of the preterm baby whilst the umbilical cord remains intact. This device has been described in detail in recent literature [6,11].

In addition, we aimed to provide DRC for all babies once stabilised, prior to moving the baby to the NICU. Once a baby was stable, with acceptable heart rate, oxygen saturation and appropriate thermal care in place, the baby was passed to the mother for a period of physical contact lasting up to 5 min. This was not skin-to-skin care. The baby continued to receive care wrapped in a plastic bag and in contact with a self heating gel mattress (Transwarmer (Drager Medical, Hemel Hempstead, UK)) if appropriate, they were also wrapped in warmed towels and a hat was in place. This period of contact was achieved for babies who were receiving respiratory support using continuous airway pressure (CPAP) (Figure 2) and in babies who were ventilated following tracheal intubation. In some families, the period of cuddling was shared with the father and following delivery by caesarean section under general anaesthetic, the cuddle was offered to the father alone if he was present.

We completed a 12-month quality improvement programme (QIP) from January 2020 to December 2020. Data were collected manually and from the electronic patient record (Badgernet full EPR, Clevermed, Edinburgh, UK) following the delivery of all babies born at less than 32 weeks’ gestation. For the purposes of this project, we used the World Health Organisation definitions [2] to classify babies into 5 temperature groups; severely hypothermic (<32 °C), moderately hypothermic (32–35.9 °C), cold stress (36–36.4 °C), normothermic (36.5–37.5 °C) or hyperthermic (>37.5 °C). Axillary temperature was measured in all babies at all times using an electronic thermometer (Welsh Allyn Suretemp Plus, Welsh Allyn, Buckinghamshire UK).

A multidisciplinary team (MDT) was assembled consisting of neonatal consultants, trainees, nurse practitioners and neonatal nurses. The proportion of babies in each temperature group was measured monthly. A detailed review was conducted for each baby who had an admission temperature below 36.5 °C. The MDT met monthly and reviewed all data including the birth location, presentation at birth and mode of delivery as well as the weight and gestation of all babies who were hypothermic or hyperthermic to try to identify common associations with these outcomes. Most importantly, interviews were conducted with the staff responsible for or involved in providing the baby’s care during stabilisation and transfer to gain their views on what they thought they could do differently in the future similar circumstances to prevent this outcome. Strategies to implement better thermoregulation practice were formulated by the MDT based on these data.

A series of Plan, Do, Study, Act (PDSA) cycles were performed using a standard methodology [12]. PDSA cycles are an iterative method of quality improvement that are used sequentially to change practice in order to improve outcomes. The basic structure of a PDSA cycle is:

Plan—to define what clinical practice should be to achieve the best outcome based on what we know about the subject. Decide what data are required to study the outcome (performance metric) and to establish a system of data collection.Do—to implement best practice and collect the data prospectively.Study—to review the data and assess performance against the metric and understand what the causes of underperformance are.Act—to make recommendations about changes in practice to improve performance and plan the next cycle.

Not all of the PDSA cycles we undertook were synchronous. Once the project was commenced (the initial P and D phases), there was a monthly review of data (S) and new actions were taken if the MDT were able to make a new recommendation (A). Each intervention was then planned (P) and implemented (D). During some meetings new cycles could be started before a previous cycle had completed (e.g., recommendations about managing twins on the Lifestart trolley were made in June, but we needed several sets of twins to be born in order to assess the effectiveness of this, in the meantime we implemented other changes in practice). Similarly, new actions were not implemented at every monthly meeting as no new changes in practice could be recommended based on the data available at that time.

Communication of changes in practice to the clinical staff required to implement them is a key part of the “Do” phase of the PDSA cycle. In this study each new strategy was communicated to the staff on the unit through a variety of methods:The performance data generated each month were displayed graphically in a bi-monthly newsletter which also summarised new practice changes implemented.‘Top tips’ posters communicating key changes in practice were produced.A range of methods of information dissemination were used to reach staff who accessed information in different ways. The newsletters and ‘Top tips’ posters were displayed across the neonatal unit, appeared on the unit’s social media pages and were emailed to all medical and nursing staff.The unit’s education team incorporated all new changes into staff induction and annual update teaching.We reinforced important messages or changes in practice using the unit’s system of “lesson of the week” announcements made at each shift handover.Written unit policies were amended to include new changes in practice as appropriate.

Some interventions also required liaison with our colleagues in other departments such as maternity and medical engineering.

The project was performed in a large women’s hospital in the UK which houses a tertiary NICU. Comparisons were made with the rates of hypothermia seen on our unit in the preceding 12 months. The statistical significance of any differences were tested using Chi-squared with Yates correction. Performance data were also collected for a 3 month period following the end of the intervention period to assess the impact of interventions made in the latter part of the period. The project was approved by the Institutional Clinical Effectiveness Senate (approval date 15 November 2019, approval number QIP 0058).

## 3. Results

130 babies less than 32 weeks’ gestation were admitted to the unit between January 2020 and December 2020. The median (range) gestation was 29 (22 to 31) completed weeks. The median (range) Birth Weight was 1225 (510 to 2810) grams.

In total 90 babies (69.2%) received DCC with 69 (53%) receiving a least 2 min of DCC and 21 (16.1%) receiving between 1 and 2 min of DCC. During this period we also successfully implemented Delivery Room Cuddles (DRC) with 79 (60%) of the babies documented to have received this.

A number of thermoregulation strategies were implemented during the project during the PDSA cycles:

Based on the performance data from 2019, the MDT made some immediate changes to practice at the start of the QIP intervention period (January 2020). These included:
○Hats for all preterm babies—We noticed that some babies who were not receiving respiratory support were not having a hat applied after birth. Subsequent routine practice included putting a hat on all babies as soon as possible.○An increase in the use of plastic bags to maintain normothermia from our previous gestation threshold of 30 weeks to 32 weeks.○An increase in the use of self heating gel mattresses to maintain normothermia from our previous gestation threshold of 28 weeks to 30 weeks.March 2020: Temperature checks—on the basis of the data, we decided to measure 3 temperatures following delivery. The first was taken as soon as practically possible after birth, the second after DRC and the third on arrival to the unit. This allowed us to understand at what point during the stabilisation and transfer period the babies were moving out of range and to target our thermoregulation interventions.May 2020: The importance of activating the self heating gel mattress well in advance of the birth was recommended as there is a time delay before the peak temperature is reached.June 2020: Introduction of the Neohelp bag (Vygon, Swindon, UK)—This is a double walled plastic bag with a hood and a Velcro opening at the front which allows the cord to remain intact whilst keeping the baby covered. This device was found to be effective and so was introduced into our regular practice for all babies born before 32 weeks.June 2020: Lifestart trolley at multiple births—we reviewed the optimal way to use a single Lifestart trolley at a multiple preterm birth. Detailed instructions for best practice was disseminated using a “Top Tips” poster (Figure 3).June 2020: (following a period of evaluation of practicability in May): Use of a non-interruptible power supply for the overhead heater during transfer to the neonatal unit. On our unit, babies are transferred from the labour ward to the neonatal unit using a Panda resuscitation platform (GE Healthcare, Buckinghamshire, UK). We found that some babies were becoming cold during transfer. The Panda can be attached to a non-interruptible power supply using a device called ‘The Shuttle’ (GE Healthcare, Buckinghamshire, UK). This allowed the overhead heater to remain on during transfer to the neonatal unit.October 2020: For babies born in theatre, we found that the transwarmer underneath the sterile drape was often left behind when transferring the baby to the resuscitation platform. Subsequently we used two transwarmers for delivery in theatre; one was placed underneath the sterile drape that covers the LifeStart whilst the baby was receiving care with an intact umbilical cord and one was placed on the Panda resuscitation platform used for further stabilisation and transfer to the neonatal unit.November 2020: Breech deliveries—the data showed that several babies born by vaginal breech delivery were hypothermic despite the above interventions. Following a discussion with the obstetric team, we agreed to cover the body of babies born as breech deliveries with the Neohelp bag whilst waiting for the head to be delivered.November 2020: Turning down the overhead heater—in babies who had a post DRC temperature of >37.5 °C, the overhead heater was reduced to 50% power prior to leaving for the neonatal unit.

The proportion of babies in each of the temperature groups during each month of the project is shown in Figure 4. The proportion of babies in each month that had hypothermia is shown in Figure 5. Assessing the individual impact of each of the new strategies that we introduced is difficult because of the time lag between each intervention and outcome and the non-synchronous nature of the PDSA cycles. The cumulative impact of the project is demonstrated in these charts which show a reduction in the rate of hypothermia during the latter part of the QIP period and into the following 3 months.

We have compared temperature outcomes from the babies admitted during the QIP intervention period with the 109 babies less than 32 weeks’ gestation born between January 2019 and December 2019. The lower number in the 2019 cohort was a consequence of normal variation in unit activity. The distribution of babies by temperature group between the two cohorts is shown in Figure 6.

During the QIP intervention period there were no cases of severe hypothermia (<32 °C), only 1 baby (0.77%) was admitted with moderate hypothermia (32–35.9 °C) and 12 babies (9.2%) were admitted with cold stress (36–36.4 °C). When compared to the 2019 cohort, there has been a significant reduction in babies admitted with a temperature below 36.5 °C from 22.3% to 10% (*p* = 0.006). This improvement was maintained after the intervention period. The rate of hypothermia in the last 6 months of data collection (last 3 months of the QIP intervention period plus the following 3 months) was 4/73 babies (5.5%). There were no hypothermic babies in the first 3 months after the QIP intervention period.

There was a slight increase in the rate of admission hyperthermia from 14.7% (20/109) in 2019 to 15.4% (20/130) in 2020, although this increase was not statistically significant (*p* = 0.88). During the 3 months after the end of the QIP intervention period 3/32 (9.4%) of babies were hyperthermic.

We were interested to understand whether the hyperthermia was due to iatrogenic warming, or due to the prevention of heat loss in already pyrexial babies. We had introduced a policy of making multiple temperature measurements at specific times during the stabilisation and transfer period during the study as described above. There were 67 babies who had paired temperature measurements on the labour ward and on admission to the neonatal unit. 55 of these were normothermic on admission and 12 were hyperthermic. Babies who were hyperthermic on admission had higher temperatures measured during their stabilisation on labour ward (Figure 7). The median (range) temperature on labour ward of the babies who were hyperthermic on admission was 37.5 °C (36.9 °C to 38.7 °C), compared to 36.9 °C (36.1 °C to 38.2 °C) in the babies who were normothermic on admission (*p* = 0.0004). All but one of the babies who were hyperthermic on admission to the neonatal unit had a temperature on labour ward above 37 °C.

## 4. Discussion

Admission hypothermia is preventable by improved care, both in the delivery room and during the transfer to the neonatal unit. Our results have demonstrated that it is possible to almost eradicate significant admission hypothermia in a busy NICU. This is achievable whilst still practising DCC and DRC, both of which are beneficial in the immediate care of the newly born preterm baby.

The solutions that we have developed in our QIP may not be the best solutions in another environment and we recommend that other units adopt a similar approach to develop their own best solution. Our solution may provide a useful starting point however and we have included a “Step by Step guide” (Figure 8) which illustrates our learning during this project.

Ambient temperature was not recorded systematically during our project, so we are unable to comment on its role in hypothermia in our own population. Published standards state that a temperature of 23 °C to 25 °C should be achieved for the delivery room [13].

A systematic review of trials of deferred cord clamping found little evidence of an impact on admission temperature [5] so we did not expect to see the increase in admission hypothermia that we experienced when we implemented DCC into our practice. Further examination of the published admission temperature data from 7 trials in the systematic review however shows that, although there was little or no impact of DCC on admission temperature, the rate of hypothermia in both intervention and control groups was high. These data are summarised in Table 1. Data are reported as mean (SD) in each trial and the reported mean is below 36.7 in 5 of the 7 trials, with wide standard deviations, demonstrating that a significant proportion of the participants in both the intervention group and the control groups of these trials, were hypothermic on admission. The mean (SD) temperature on admission in our cohort was 37 (0.53) °C. Even in one of the trials reporting mean admission temperatures of above 36.5 °C [7], there were still 11.6% of babies with an admission temperature below 36 °C. Taken together, these observations suggest that in clinical areas where there are already moderate rates of admission hypothermia, the introduction of DCC makes little difference to these rates. Our experience, in a unit with a low rate of hypothermia, is that the introduction of DCC was associated with an increase in hypothermia.

Despite the significant reductions in hypothermia we achieved, we failed to achieve the NNAP standard of 90% of babies in the normothermic range during the QIP intervention period, although that was achieved in one of the subsequent 3 months (Figure 9). During the QIP intervention period, the commonest cause of failing to meet the NNAP standard was hyperthermia (*n* = 20) rather than hypothermia (*n* = 13). Our exploration of the data, presented above, shows that much, if not all, of the admission hyperthermia we saw was a consequence of prevention of heat loss in already pyrexial babies rather than iatrogenic warming.

Although the evidence for a causal relationship between hypothermia and adverse outcome at preterm birth is well established and accepted, the evidence relating to hyperthermia is not compelling. In fact, there is little published evidence in relation to this matter. Lyu et al. found an increased rate of a composite adverse outcome (severe neurological injury, severe retinopathy of prematurity, necrotising enterocolitis, bronchopulmonary dysplasia, and nosocomial infection) in babies with an admission temperature above 38 °C [20]. Sharma et al. also reported an increase in a composite adverse outcome (mortality or major morbidity) in babies with admission temperature >37.5 °C [21]. The Clinical Risk Indicator for Babies II score (CRIB II) [22] assigns an increased score (increased risk of mortality) to babies with an admission temperature >37.5 °C. We assume that this is because the dataset on which the score was developed found an increased risk of death in those babies, although this is not explicitly stated in the paper describing the development of the score. Although these studies provide some evidence of an association between hyperthermia on admission and adverse outcome in preterm babies, none of the published studies has established a causal link and it is possible that this association is due a shared aetiology with chorioamnionitis, which also has a strong association with preterm brain injury [23,24]. The 2020 International Liaison Committee on Resuscitation recommendations for neonatal life support include a weak recommendation, based on low certainty evidence, that hyperthermia (greater than 38 °C) should be avoided due to the potential associated risks [13]. On this basis we believe that a small increase in hyperthermia is probably an acceptable consequence of a reduction in hypothermia. Nonetheless, we still strive to avoid hyperthermia by switching off the power supply to the overhead heather during transfer of babies with a temperature following DRC above 37 °C as described above.

## 5. Conclusions

This project highlights the importance of the perinatal MDT for driving change in a large complex organisation. Collaboration and clear communication with our obstetric, midwifery, medical engineering and estate staff colleagues were vital to facilitating the success of this project. We have used this approach previously to successfully drive change in our unit [8,25]. The team approach (rather than a top-down approach) promotes ownership of the project, generates enthusiasm and helps create a genuine desire for success.

## Figures and Tables

**Figure 1 children-09-00075-f001:**
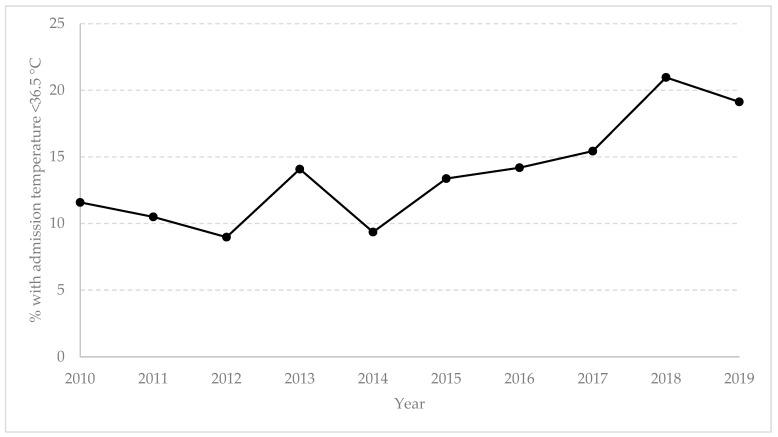
Proportion of babies born before 32 weeks gestation with an admission temperature below 36.5 °C across time during the introduction of DCC on our unit.

**Figure 2 children-09-00075-f002:**
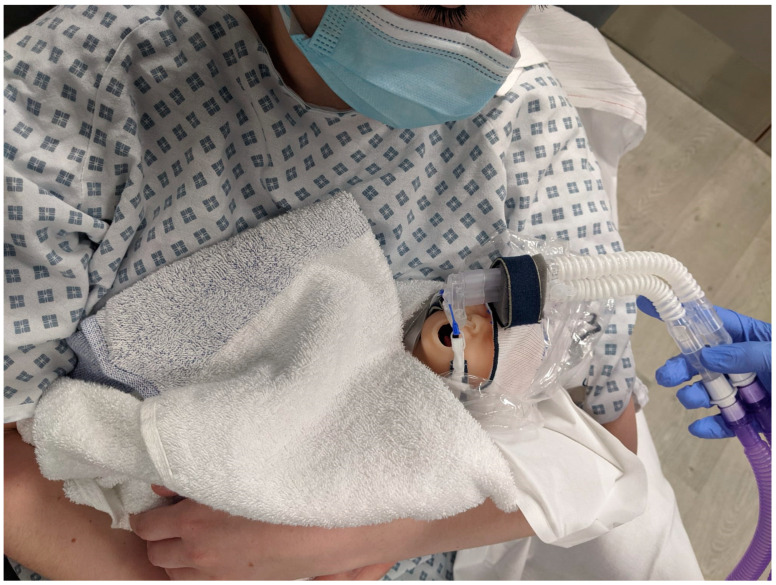
Simulation of a preterm baby receiving Delivery Room Cuddles whilst receiving respiratory support with nasal CPAP.

**Figure 3 children-09-00075-f003:**
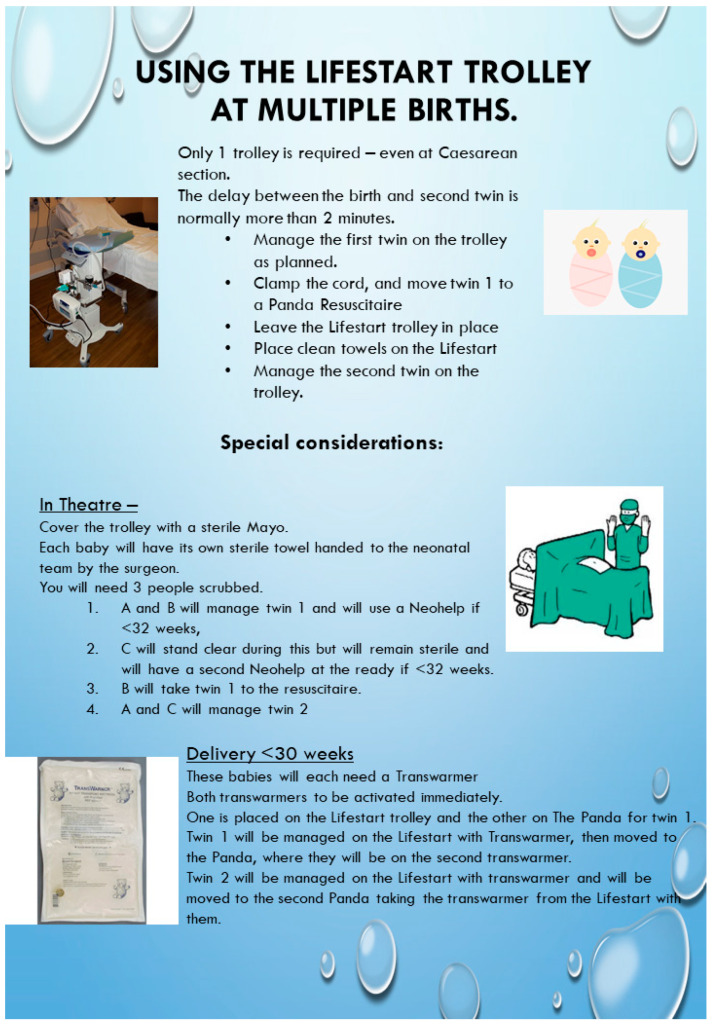
An example of a “Top Tips” poster used to communicate best practice to staff.

**Figure 4 children-09-00075-f004:**
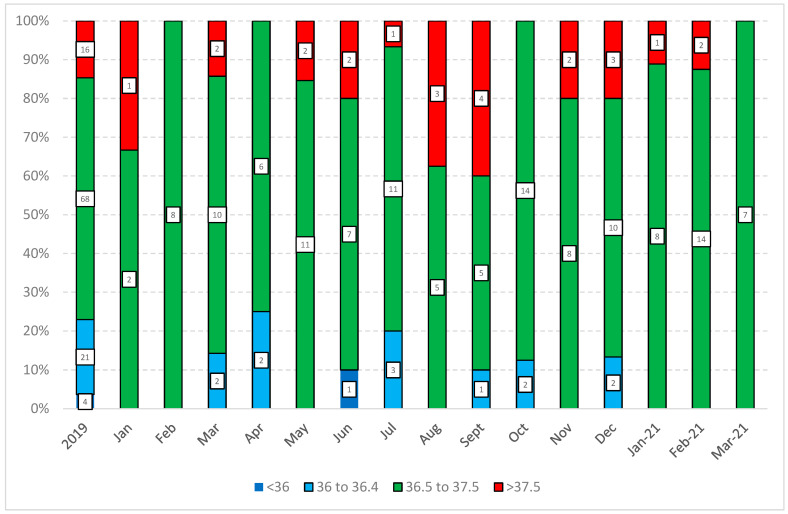
The proportion of babies admitted in each temperature category during 2019, each month of the QIP intervention period and during the 3 month period after the QIP intervention period. Numbers on the bars are absolute numbers in each category in each month.

**Figure 5 children-09-00075-f005:**
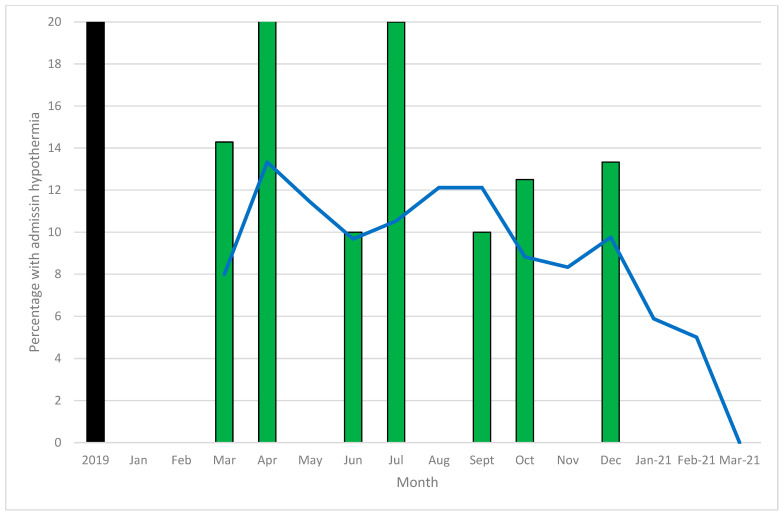
The percentage of babies with admission hypothermia in 2019 (black bar), during each month of the QIP intervention period (green bars) and during the 3 month period after the QIP intervention period (there were no hypothermic babies during this period). The blue line represents a 3 month rolling average.

**Figure 6 children-09-00075-f006:**
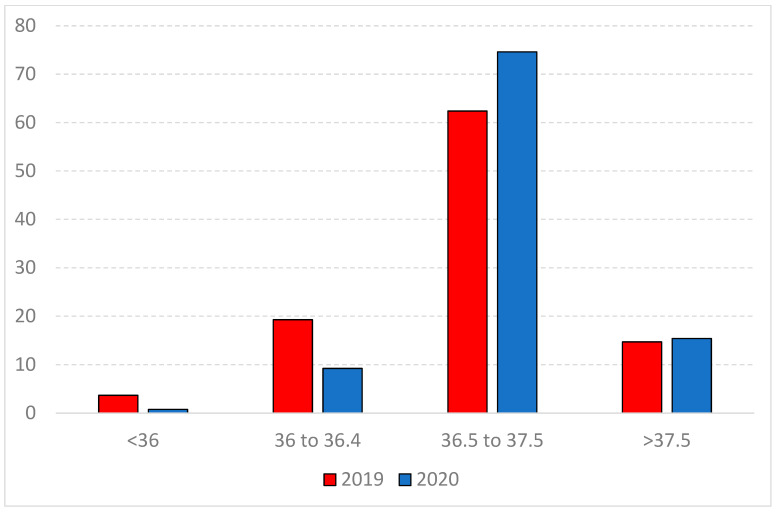
The distribution of babies between the different temperature categories in the 2019 cohort and the 2020 cohort.

**Figure 7 children-09-00075-f007:**
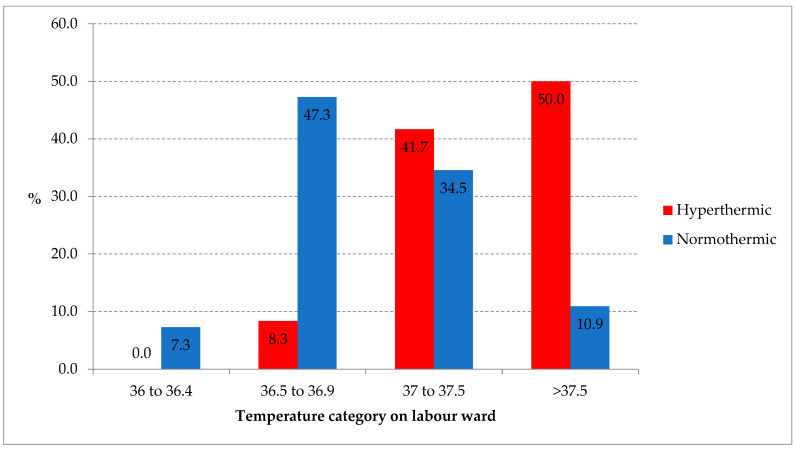
Temperature category at birth for babies who were normothermic on the labour ward (blue bars) and babies who were hyperthermic on the labour ward (red bars).

**Figure 8 children-09-00075-f008:**
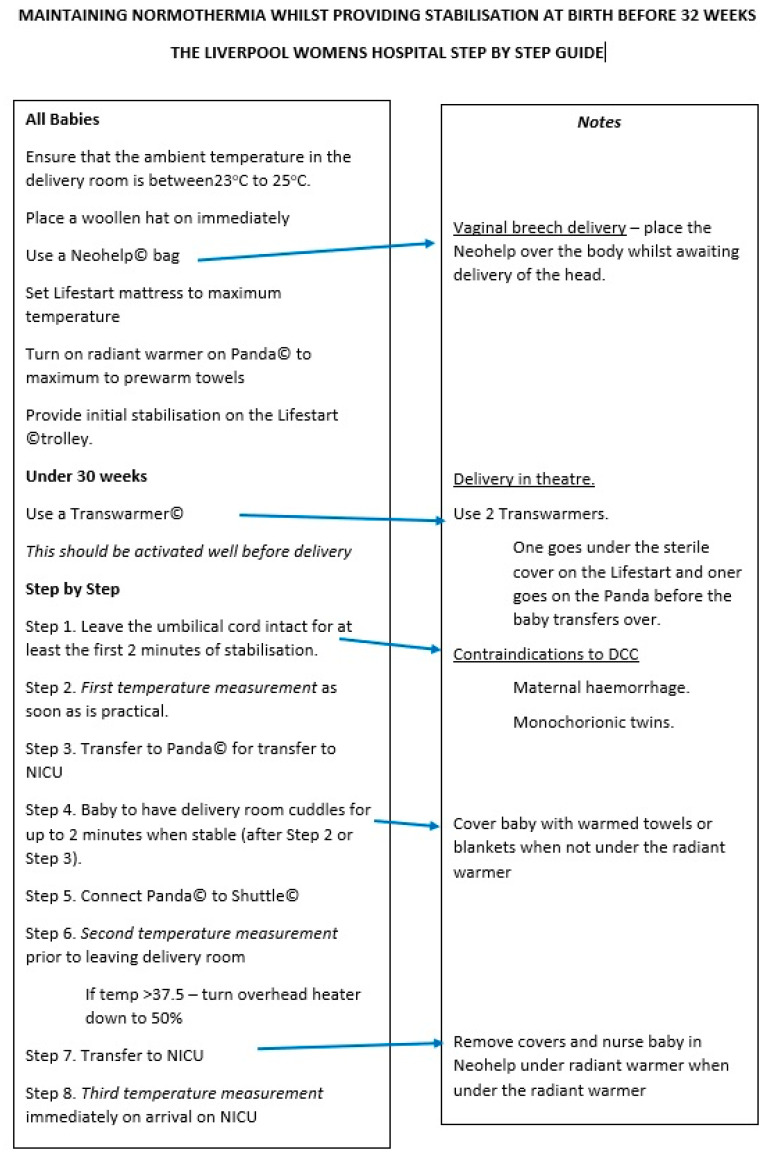
A Step by Step Guide to maintaining normothermia during preterm stabilisation with an intact umbilical cord, summarising all of our learning during this project.

**Figure 9 children-09-00075-f009:**
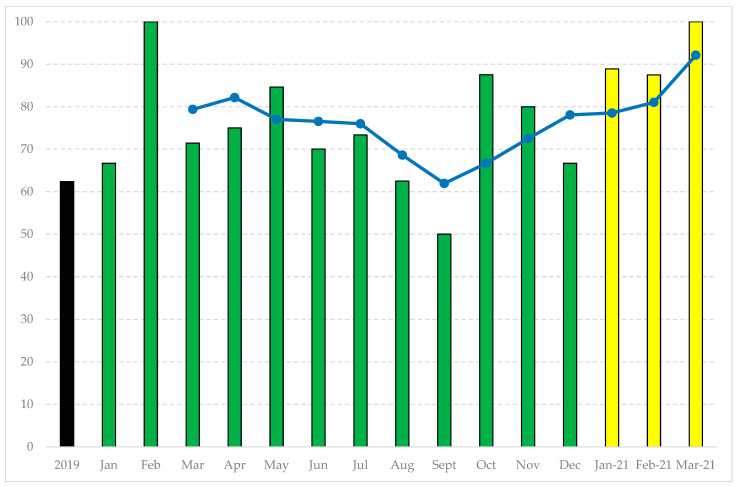
Compliance with the NNAP standard by month during 2019 (black bar), each month during the QIP intervention period (green bars) and each of the 3 months immediately after the QIP intervention period (yellow bars). The blue line represents a 3 month rolling average.

**Table 1 children-09-00075-t001:** Published admission temperature data from randomized controlled trials of deferred cord clamping at preterm birth.

		Group Temperature Mean (SD) °C		
Study	Subjects	Control	Intervention	Comment
Mercer [14]	32	36.3 (17.2)	36.3 (17.2)	Minimum temperatures reported were 35.2 °C in Control group and 34.7 °C in the intervention group
Mercer [15]	72	36 (0.8)	36.2 (6)	Minimum temperature reported were 33.8 °C in the control group and 34.4 °C in the intervention group
Backes [16]	40	35.7 (16.8)	36.3 (16.8)	
Dipak [17]	53	34 (0.7)	33.9 (0.8)	
Tarnow Mordi [18]	1248	36.4 (0.9)	36.3 (0.8)	
Duley [7]	266	36.9 (0.6)	36.7 (0.6)	31 (11.6%) trial subjects had admission temp <36 °C
Yunis [19]	60	36.8 (0.4)	36.7 (0.5)	

## Data Availability

The data presented in this study are available in the text and tables of this manuscript.

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
