# Peer review of "Maintaining Normothermia in Preterm Babies during Stabilisation with an Intact Umbilical Cord"

_children, 2022, doi:10.3390/children9010075_

Round 1

Reviewer 1 Report

The authors have described the results of an important QI study aimed at improving hypothermia rates while preserving performance of DCC. I have a few comments to help improve the flow and clarity of the paper.

  • Some short description of a standard way to perform DRC in addition to the references provided would be helpful.
  • Further description of the factors assessed during data review would allow readers to better understand the rigor of the methodology.
  • A brief description of the changes implemented with each PDSA cycle with how many PDSA cycles occurred is needed. Some thermoregulatory interventions are listed in the results, would move that to methods with a description of which occurred during which PDSA cycle and when (what month) each PDSA cycle occurred. Would be of interest given that the authors break down which temperature range babies fall into by month.
  • In results, the median with range listed is full range or IQR?
  • Some brief description of results in Figures 2&3 in the results text would be beneficial.
  • Explain why in figure 1 the authors think the baseline rate of hypothermia appears to be trending up in the 2-3 years prior to implementation of DCC.
  • Figure 3 is unclear. Are the numbers overlaying the bars the absolute numbers or percentages? Would possibly be more clear if displayed as a clustered bar graph or simply if numbers were removed.
  • In the discussion, would include references for hyperthermia being less clinically significant than hypothermia.

Author Response

I have submitted my response to the reviewers in the attached document

Reviewer 2 Report

This is a solid manuscript describing a quality improvement project in a NICU and I especially appreciate the honesty of the conclusion that introduction of DCC lead to an increase in hypothermia. I will just point out the very few things that need a bit of improvement or clarification:

  • Starting at row 86, the authors mention searching for different strategies based on the individual experience and number of occurred cases – were those strategies implemented during the project? If so, the results could have been a lot better at the end of the studied period compared to the beginning, but they are comparable
  • There is no mention of the ambient temperature – was it different in delivery rooms compared to operating theatres? Did it have any influence on the infant’s body temperature?
  • Are Delivery Room Cuddles limited to the delivery rooms? Do those involve skin-to-skin contact or just touching while the baby is on the radiant warmer or in the Neohelp bag?
  • Was admission temperature correlated to the necessary period of stabilization?
  • In rows 146 and 148, authors should place the appropriate symbol for degree (Ëš) where needed
  • On row 156, I suggest switching to “a self-heating gel mattress – Transwarmer (Draeger, UK)”
  • Row 168 – please specify how the temperature was measured – skin or intrarectal. Was it performed the same way for all the infants?
  • I would move up measure number 8 and demote numbers 6 and 7, as they tend to more peculiar situations, not the majority of cases
  • For Figure 1, there is discordance between the legend on the left side and the title of the figure, concerning the temperature limit it depicts
  • Percentages should be added to Figure 2
  • Something should be commented about the results from August and September when, according to figures 3 and 5, there was the least compliance with the NNAP, due to hyperthermia. Was this issue further investigated, maybe linked to some particular procedures, or some members of the neonatal team?
  • Titles 3, 5, 11 from the reference list should be rewritten for conformity.

Author Response

I have responded to the reviewer's comments in the attached document
